# Effects of Zirconium Silicide on the Vulcanization, Mechanical and Ablation Resistance Properties of Ceramifiable Silicone Rubber Composites

**DOI:** 10.3390/polym12020496

**Published:** 2020-02-24

**Authors:** Jiuqiang Song, Zhixiong Huang, Yan Qin, Honghua Wang, Minxian Shi

**Affiliations:** School of Materials Science and Engineering, Wuhan University of Technology, Wuhan 430070, China; songjq9@126.com (J.S.); zhixiongh@whut.edu.cn (Z.H.); qinyan@whut.edu.cn (Y.Q.); 13343434177@163.com (H.W.)

**Keywords:** ceramifiable silicone rubber composite, zirconium silicide, vulcanization, mechanical strength, ablation resistance

## Abstract

Ceramifiable silicone rubber composites play important roles in the field of thermal protection systems (TPS) for rocket motor cases due to their advantages. Ceramifiable silicone rubber composites filled with different contents of ZrSi_2_ were prepared in this paper. The fffects of ZrSi_2_ on the vulcanization, mechanical and ablation resistance properties of the composites were also investigated. The results showed that the introduction of ZrSi_2_ decreased the vulcanization time of silicone rubber. FTIR spectra showed that ZrSi_2_ did not participate in reactions of the functional groups of silicone rubber. With the increasing content of ZrSi_2_, the tensile strength increased first and then decreased. The elongation at break decreased and the permanent deformation increased gradually. The thermal conductivity of the composite increased from 0.553 W/(m·K) to 0.694 W/(m·K) as the content of the ZrSi_2_ increased from 0 to 40 phr. In addition, the thermal conductivity of the composite decreased with the increase of temperature. Moreover, thermal analysis showed that the addition of ZrSi_2_ increased the initial decomposition temperature of the composite, but had little effect on the peak decomposition temperature in nitrogen. However, the thermal decomposition temperature of the composite in air was lower than that in nitrogen. The addition of ZrSi_2_ decreased the linear and mass ablation rate, which improved the ablative resistance of the composite. With the ZrSi_2_ content of 30 phr, the linear and mass ablation rate were 0.041 mm/s and 0.029 g/s, decreasing by 57.5% and 46.3% compared with the composite without ZrSi_2_, respectively. Consequently, the ceramifiable silicone rubber composite filled with ZrSi_2_ is very promising for TPS.

## 1. Introduction

Ceramifiable polymer composites play important roles in the field of thermal protection systems (TPS) for rocket motor cases, due to their advantages of low density, low heat conduction, excellent mechanical properties and ablation resistance [1,2]. The ceramifiable polymer composites for TPS include fiber reinforced phenolic resin composites [3,4,5], ceramifiable EPDM composites [6,7,8,9] and ceramifiable silicone rubber composites [10,11,12]. During the process of ablation, the residue of decomposed silicone rubber is loose and weak, and can be peeled off from the ablation surface under the erosion of high-speed airflow, resulting in deeper ablation. Therefore, in order to improve the ablation resistance of silicone rubber composites, it is necessary to add various functional fillers to the composites. The fillers undergo ceramifying transformation and form a ceramic-like structure protective layer with certain mechanical strength to reduce the ablation rate during the ablation process. Aluminosilicate minerals, metal and nonmetal oxides, carbides, borides, nitrides, graphite, etc. are usually included as ceramifiable fillers.

Cheng et al. [13,14,15,16] used frits with low melting point and mica in ceramifiable silicone rubber composites firstly and systematically studied the ceramifying transformation of aluminosilicate minerals during the combustion process. The frits melted into a liquid phase at a low temperature, which played a role in bonding the decomposition products of silicone rubber and other fillers, providing conditions for the sintering reaction of mica and other fillers. Yang [17] studied the effect of ZrO_2_ and ZrC powders on the properties of ablative silicone rubber composites. It was found that both powders increased the tensile strength of the composites, but decreased the elongation at break. After ablation by oxyacetylene flame, the ablation rate of the composites decreased significantly. In addition, the effect of ZrO_2_ on the ablation rate of the composites was more obvious. The linear ablation rate and mass ablation rate were 0.021 mm/s and 0.036 g/s when the amount of ZrO_2_ was 40 phr, respectively. Anyszka [18] and Zhang [19] studied the properties of the ceramifiable silicone rubber composites filled with surface modified montmorillonite. The surface organic modification of montmorillonite was helpful to the formation of nanoporous structures during the ceramifying reaction process, which could improve the thermal insulation performance of the ceramic layer without effectively reducing the mechanical strength. Zhang and his colleagues [20] also studied the effects of graphene nanoplatelets on the properties of ceramifiable silicon rubber composites. After organic modification, graphene nanoplatelets had a good combination with silicone rubber, which improved the pyrolysis temperature of silicon rubber by nearly 40 °C, and promoted the carbonization reaction. The linear ablation rate and mass ablation rate were reduced by 39% and 27%, respectively. In addition, further micro-structure research showed that the addition of nanoplatelets led to the growth of SiC nanowires during the process of carbonization, improving the mechanical strength of the carbonized layer. Liu [21] studied the effects of SiC, HfO_2_, ZrO_2_ and ZrB_2_ on the ablation resistance of high temperature vulcanized silicone rubber (HVSR) composite. The linear ablation rates of ZrB_2_/HVSR, ZrO_2_/HVSR, HfO_2_/HVSR, SiC/ HVSR were 0.152, 0.161, 0.219 and 0.207 mm/s, respectively. The back temperature of ZrB_2_/HVSR was the smallest in the ablation process, and the peak back temperature was only 77 °C, which had good heat insulation performance. Moreover, the surface structure after ablation was more compact, and the holes were smaller, which resisted the erosion of high-speed particle airflow effectively. All of the above ablative fillers have high density. The introduction of the fillers into the ceramifiable silicone rubber composites will increase the density of the composites, which brings great difficulty to the lightweight of the composite. Furthermore, the aluminosilicate minerals with relatively low density have poor ablation resistance and cannot operate for a long time in the extreme environment above 2000 °C, so it is necessary to find and develop more suitable ceramifiable fillers. 

Zirconium silicate (ZrSi_2_) is a kind of silicide ceramic material with a high melting point, and adensity of 4.88 g/cm^3^. However, the densities of zirconium carbide (ZrC) and zirconia (ZrO_2_) are 6.73 g/cm^3^ and 5.85 g/cm^3^, respectively. ZrSi_2_ has relatively less effect on the density of the composites. SiO_2_, ZrO_2_ and other ceramic products with high hardness and mechanical strength will be formed in the process of ablation. Ding [22,23] introduced ZrSi_2_ into C-phenolic ablative composites and found that the linear and mass ablative rates of the composites were reduced by 80.5% and 55.2%, respectively. A SiO_2_-ZrO_2_ ceramic protective layer was formed during the ablation process, which effectively insulated the erosion of internal materials from heat flow and oxygen. In addition, the introduction of ZrSi_2_ improved the thermal stability of phenolic resin, but had no obvious effect on the thermal curing process of phenolic resin.

In this paper, ZrSi_2_ was added to ceramifiable silicone rubber composite. The effects of ZrSi_2_ content on the mechanical property, thermal stability, vulcanization and ablation resistance of the composite were studied. The vulcanization, thermal decomposition and ceramifying process of ceramifiable silicone rubber composite filled were with ZrSi_2_ analyzed.

## 2. Experimental

### 2.1. Materials

Methyl vinyl silicone rubber (SR) with the average molecular weight of 600,000 and the vinyl content of 0.18% per mole was provided by Chengdu Zhonghao Chenguang Technology Co. Ltd. (Chengdu, China). Fumed silica (SiO_2_, Shanghai Jingchun Bio-Chem Technology Co. Ltd., Shanghai, China) with a BET surface area of 200 m^2^/g was used to improve the mechanical performance of the composite. Zirconium silicate (ZrSi_2_, Shanghai Naiou Nano Technology Co. Ltd., Shanghai, China) with an average particle size of 10 μm was used as ceramifiable filler. Graphite (C) with an average particle size of 1 μm was purchased from Shanghai Jingchun Bio-Chem Technology Co. Ltd. 2,5-Dimethyl-2,5-di(*tert*-butylperoxy)hexane (DBPH) and cyclohexane were also purchased from Shanghai Jingchun Bio-Chem Technology Co. Ltd. (Shanghai, China), and used as the curing agent and swelling agent, respectively. 

### 2.2. Preparation of the Ceramifiable Silicone Rubber Composites

Firstly, ZrSi_2_ powder was put into the oven and dried at 105 °C for 24 h. Then, the ingredients were mixed on a clean two-roll mill with a gear ratio of 1:1.2 at room temperature. The silicone rubber was first softened, and then fumed silica, graphite, and ZrSi_2_ were added until a homogeneous batch was obtained. Finally, the curing agent DBPH was added and processed until a visually good dispersion was achieved. The total mixing time was about 30 min. The samples were molded to platens by press vulcanizer at 170 °C with pressure of 10 MPa for 15 min. Then, the platens were put into an oven at 200 °C to air-dry for 4 hours for additional vulcanization. The formulations of silicon rubber composites are given in Table 1.

### 2.3. Characterization

#### 2.3.1. Vulcanization Characteristics

The vulcanization curve of the ceramifiable silicone rubber composite with different content of ZrSi_2_ was performed by a moving die rheometer (Kaiyuan Test Machinery Factory, Kaiyuan, China) according to Chinese standard GB/T 9869-2014. The dimensions of the circular samples were 40 mm (diameter) × 4 mm (thickness). The temperature and time were 170 °C and 12 min, respectively.

ATR-FTIR spectroscopy was obtained in the range of 400–4000 cm^−1^ at a resolution of 1 cm^−1^ on a Nexus FTIR spectrophotometer (Thermo Nicolet, Waltham, MA, USA) for samples with different content of ZrSi_2_ to determine whether ZrSi_2_ participated in the vulcanization of silicone rubber.

#### 2.3.2. Mechanical and Physical Properties

The tensile strength and elongation at break tests of the ceramifiable silicone rubber composite with different content of ZrSi_2_ were performed using a universal testing machine (Instron-4465, Instron Engineering Corporation, Norwood, MA, USA) according to Chinese standard GB/T 528-2009. The loading speed was 500 mm/min. The dimensions of the dumbbell samples were 115 mm (length) × 6 mm (width of narrow area) × 2 mm (thickness).

Thermal conductivity of the composite was measured by a thermal conductivity measuring apparatus (QTM-500, Kyoto Electronic Industry Co., Ltd, Kyoto, Japan) at different temperatures. The dimension of the samples was 120 mm (length) × 100 mm (width) × 50 mm (thickness). The results were also the average value of three specimens.

#### 2.3.3. Thermal Gravimetric Analysis

Thermal gravimetric analysis (TGA, STA449C/3/G, NETZSCH, Selb, Germany) was conducted to investigate the thermal stability of the samples with different contents of ZrSi_2_ under air and N_2_. Then, a series of samples were heated at a rate of 10 °C/min. The relative mass loss and decomposition temperatures of the samples were recorded from room temperature to 1000 °C.

#### 2.3.4. Ablation Test

The ablation performance of the composites was evaluated by oxyacetylene torch test. The vertical distance of the nozzle to the sample was 10 mm and the inner diameter of the nozzle was 2.0 mm. The flow rates of oxygen and acetylene were 1512 L/h and 1116 L/h, respectively. Time of ablation was 20 s. The linear ($R_{l}$) and mass ($R_{m}$) ablation rates were calculated according to the following formulae:(1)$$R_{l}=-\frac{d_{2}-d_{1}}{T},$$
(2)$$R_{m}=-\frac{m_{2}-m_{1}}{T},$$
where $d_{1}$, $d_{2}$, and $m_{1}$, $m_{2}$ are the thickness and mass of the sample before and after ablation test, respectively, and T is the testing time. The $d_{2}$ thickness was measured after removing the char layer.

#### 2.3.5. Morphology

The morphology of the residue after ablation was characterized by field emission scanning electron microscopy (FESEM, Zeiss Ultra Plus, Carl Zeiss NTS GmbH, Oberkochen, Germany).

#### 2.3.6. X-ray Diffraction Analysis

The crystal phases of the residue were identified by an X-ray diffraction (XRD) with a D8 ADVANCE diffractometer (Bruker, Billerica, MA, USA) with Cu Kα (*λ* = 0.1542 nm) radiation at a generator voltage of 40 kV and a generator current of 400 mA. The scan was conducted from a 2*θ* angle of 5 to 80° with a step interval of 4°.

## 3. Results and Discussion

### 3.1. Vulcanization Characteristics

In order to study the effect of ZrSi_2_ on the vulcanization of silicone rubber, vulcanization curves of the composite with different content of ZrSi_2_ were plotted. Figure 1 shows the torque change of the ceramifiable silicone rubber composites filled with ZrSi_2_ powder during vulcanization at 170 °C, and Table 2 lists the scorch time (*T*_10_), the positive vulcanization time (*T*_90_) and the torque at 12 min. As seen in Figure 1, the addition of ZrSi_2_ decreased the vulcanization time of silicone rubber, promoting the formation of vulcanization network structure of silicone rubber. With the increased content of ZrSi_2_, the torque increased. The addition of ZrSi_2_ increased the deformation resistance of silicone rubber, which required external force for the same deformation. In addition, from the Table 2, the torque of the ceramifiable silicone rubber composite increased with the increase of ZrSi_2_ content and the positive vulcanization time (*T*_90_) decreased from 421.80 s to 379.80 s, but there was little effect on the scorch time (*T*_10_). It was also confirmed that ZrSi_2_ could improve the mechanical properties of the ceramifiable silicone rubber composite.

Figure 2 shows the FTIR spectra of the ceramifiable silicone rubber composites filled with different contents of ZrSi_2_. The addition of ZrSi_2_ and the change of its content had little effect on the functional groups during the curing process of silicone rubber. The FTIR peaks marked at 2963 cm^−1^,1412 cm^−1^ and 1259 cm^−1^ were characteristic absorption peaks of the C–H bond in Si–CH_3_. Because the silicone rubber used in the experiment was methyl vinyl silicone rubber, there were lots of methyl on the side chains. The broad absorption peak observed at 1077 cm^−1^ was the characteristic peak of Si-O bond existed in the main molecular chain of silicone rubber. The absorption peak marked at 793 cm^−1^ was associated with the characteristic peak of Si–C bond. The FTIR spectra of the ceramifiable silicone rubber composites with different ZrSi_2_ content were compared, which illustrated that the addition of ZrSi_2_ did not produce a new characteristic absorption peak. So there was physical interaction between ZrSi_2_ particles and silicone rubber, which did not change the molecular structure of silicone rubber.

### 3.2. Mechanical and Physical Properties 

For the flexible TPS composites, some deformation will be happened during its service, such as thermal expansion of metal shell, tensile and compressive stress of solid motor grain, etc. Therefore, sufficient tensile strength and elongation are required for TPS composites. The tensile strength and the elongation at break of the ceramifiable silicone rubber composites with different content of ZrSi_2_ are showed in Figure 3. The results showed that the introduction of ZrSi_2_ reduced the tensile strength and elongation at break of the ceramifiable silicone rubber composite. With the increase of ZrSi_2_ content, the tensile strength of the composite increased first and then decreased, while the elongation at break decreased all the time. The tensile strength and the elongation at break of the composite with 30 phr ZrSi_2_ were 5.08 MPa and 364.3%, respectively. The mechanical strength and the elongation at break of pure silicone rubber are very low. After being reinforced by fumed silica, the tensile strength can reach more than 6 MPa, and the elongation is more than 500%. The addition of ZrSi_2_ powder reduced the mechanical strength of the silicone rubber composites obviously. The powder destroyed the continuity of the silicone rubber matrix, resulting in local defects, which was easy to become the stress concentration point under the action of external load. With the increase of ZrSi_2_ content, the movement of silicone rubber molecular chain was hindered, which made the fracture of silicone rubber molecular chain require more energy. Thus, the tensile strength of the composite was increased. However, the effect of exceeding addition on the matrix was more than that of strength enhancement, which led to the decrease of strength. From the Figure 3, the permanent deformation at break of ceramifiable silicone rubber composite increased due to the hindrance of ZrSi_2_ to the movement of silicone rubber molecular chain. The resilience and recoverability of silicone rubber were reduced after breaking. The best additive content of ZrSi_2_ was 30 phr from the results of tensile properties.

The ceramifiable silicone rubber composite used in the field of aircraft TPS has higher requirements for thermal conductivity. The results of thermal conductivity are shown in Figure 4 and Figure 5. After adding 40 phr ZrSi_2_, the thermal conductivity of the composite increased from 0.553 W/(m·K) to 0.694 W/(m·K). Compared with polymer, ZrSi_2_ is a kind of ceramic material with high thermal conductivity. The thermal conductivity factor was added after being introduced into silicone rubber. With the increase of the content, the particles contacted with each other, and the number of contact points increased, forming heat transfer channels, which led to the increase of the thermal conductivity of the composite. Reducing the volume of the internal space of the composite resulted in the increase of the thermal conductivity indirectly. However, the thermal conductivity of the composite prepared in this paper was still at a low level in the TPS materials compared with metal, ceramic (>10 W/(m·K)) and C/C composites (>0.8 W/(m·K) [22]), meeting the current requirements for internal insulation materials. Figure 5 showed the thermal conductivity of the composite at different temperatures. The thermal conductivity of the composite decreased with the increase of temperature. In the process of heating up, the composite material had a certain degree of thermal expansion, which increased the distance between the internal particles. Moreover, the molecular chain of silicone rubber also had a certain extension, which was no longer as dense as that at room temperature, destroying the thermal conduction channels, co the thermal conductivity of the composite was reduced.

### 3.3. Thermal Stability of the Composite

Figure 6 shows the TG and DTG curves of the ceramifiable silicone rubber composite with different content of ZrSi_2_, graphite and ZrSi_2_ in nitrogen, and Table 3 lists the characteristic parameters of the thermal decomposition of the composite. *T*_5_ and *T_max_* are the temperatures of 5% weight loss and peak degradation, respectively. It can be seen that the ceramifiable silicone rubber composite containing ZrSi_2_ underwent three thermal degradation process stages. The first process at about 300–500 °C was attributed to volatilization of some small molecules and gases yielded by the decomposition of silicone rubber. At this stage, the mass loss and the decomposition rate of the composite were small, which maintained the original state of the composite basically. The second stage was at about 500–750 °C, in which the mass of the composite changed obviously, and the silicone rubber decomposed violently. Some studies have shown that fracture and rearrangement of molecular chains happened during the decomposition of silicone rubber, producing a large number of methane, small molecule oligocyclosiloxane, CO and other gaseous substances [17], which causing major mass loss. With the increase of temperature, when the temperature was above 750 °C, the thermal decomposition of the composite was basically over, and the mass of the residue was no longer changed, which was the third stage. From the Figure 6 and Table 3, the addition of ZrSi_2_ lowered the thermal stability slightly, resulting in reduction of *T*_5_ and *T_max_*. The catalytic process of silicone rubber composite by metal ions from decomposition has already been reported in the literature. But with the increasing content of ZrSi_2_, *T*_5_ increased from 429.9 °C to 490.2 °C and *T_max_* shifted from 648.9 °C to 644.7 °C. The effect of ZrSi_2_ on molecular structure was small, which had little effect on the *T_max_*. Therefore, ZrSi_2_ had little effect on the high temperature stability of the ceramifiable silicone rubber composite.

According to the characteristic decomposition data of the composite in Table 3, the influence of ZrSi_2_ on the yield of pyrolysis products of the composite at 800 °C was further analyzed. It can be seen from Figure 6 that graphite powder and ZrSi_2_ were very stable in nitrogen, and the mass change was very small with the increase of temperature. The mass loss of the composite during pyrolysis was mainly caused by the cracking of silicone rubber. Based on the pyrolysis yield of the composite without ZrSi_2_, the pyrolysis yield of the silicone rubber composite with different ZrSi_2_ content was calculated according to Equation (3). The calculation results are shown in Table 3, and there was no significant difference between the calculated values and the experimental values. Therefore, the main mass loss of the thermal decomposition of silicone rubber composite filled with ZrSi_2_ was caused by the cracking of silicone rubber, and the introduction of ZrSi_2_ did not change the pyrolysis yield of the composite in nitrogen:(3)$$R=\frac{m_{SRZ0}\times 41.35\%+m_{ZrSi_{2}}\times 99.32\%}{m_{SRZX}}\times 100\%,$$
where R is the yield of residue of composite at 800 °C. $m_{SRZ0}$ is the mass of sample SRZ0. $m_{ZrSi_{2}}$ is the amount of ZrSi_2_ in the composite. $m_{SRZX}$ is mass of the sample.

Figure 7 shows the TG and DTG curves of the ceramifiable silicone rubber composites with different content of ZrSi_2_, graphite and ZrSi_2_ in air, and Table 4 lists the characteristic parameters of the thermal decomposition of the composite. It can be seen from Figure 7 that the thermal decomposition of the ceramifiable silicone rubber composite in the presence of oxygen was quite different from that in the inert atmosphere. From Table 4, it can be concluded that the initial decomposition temperature of the composite was 388.6 °C–399.7 °C, and the peak decomposition temperature was 488.0 °C–527.1 °C, which were lower than the characteristic decomposition temperature in nitrogen. Therefore, in the oxidation environment, the presence of oxygen could accelerate the thermal decomposition of silicone rubber. In addition, the amount of ZrSi_2_ had little effect on the thermal stability of the silicone rubber composite in the oxidation environment, and the initial decomposition temperature was increased by about 11.1 °C. On the other hand, the yield of thermal decomposition residue of the composite at 680 °C was higher than that in inert atmosphere, which was related to the oxidation of silicone rubber. The silicon produced by decomposition was oxidized, increasing the mass of residue. The yield of thermal decomposition residue was 60.06% with the addition of 40 phr ZrSi_2_. However, with the increase of temperature, graphite and ZrSi_2_ were oxidized, in which CO_2_ was produced by oxidation of graphite, resulting in weight loss, while ZrSi_2_ was oxidized above 850 °C, and the mass increased by 13.11% at 1000 °C. The mass change caused by graphite and ZrSi_2_ leaded to further weight loss of the composite above 700 °C. The second weight loss of the composite with 40 phr ZrSi_2_ at 1000 °C was 14.58%, and the yield of thermal decomposition residue was 45.45%.

### 3.4. Ablation Resistance

Linear and mass ablation rate are the most intuitive performance indexes to judge ae ceramifiable silicone rubber composite. The linear and mass ablation rate of composite with different ZrSi_2_ content are shown in Figure 8. With the increasing content of ZrSi_2_, the linear and mass ablation rate decreased. When the ZrSi_2_ content was 0 phr, the linear ablation rate and mass ablation rate were 0.097 mm/s and 0.054 g/s, respectively. When 10 phr ZrSi_2_ was added, the linear ablation rate and mass ablation rate of the composite decreased to 0.078 mm/s and 0.046 g/s, decreasing by 19.6% and 14.8% compared with the composite without ZrSi_2_, respectively. The linear ablation rate and mass ablation rate of the composite with 40 phr ZrSi_2_ were 0.039 mm/s and 0.029 g/s, respectively. The decrease of linear and mass ablative rate showed that the addition of ZrSi_2_ improved the ablative resistance of the ceramifiable silicone rubber composite. The increase of ZrSi_2_ content led to the increase of the density of the composite, enhancing the erosion resistance to the air flow during the ablation process, which reduced the linear and mass ablation rate of the composite. In addition, it can be also seen from the results of thermal analysis in air that a large amount of thermal decomposition gas was released, and the weight increased at the same time, which reduced the mass ablation rate during the ablation process. According to the results of tensile strength, when the best addition amount of ZrSi_2_ was 30 phr, the linear and mass ablation rate of the composite were 0.041 mm/s and 0.029 g/s, decreasing by 57.5% and 46.3% compared with the composite without ZrSi_2_, respectively.

Digital photos of the ablated samples are shown in Figure 9. After ablation by oxyacetylene flame, lots of white residue was produced on the surface from Figure 9a. Expansions on the edge of ablation area were observed. In addition, a multilayer ablation structure that can be divided into three parts was revealed from Figure 9b. From the surface to the un-erosion material, there were ceramic layer (Ⅰ), pyrolysis layer (Ⅱ) and virgin layer (Ⅲ). And there was no obvious boundary between the layers. The thickness of pyrolysis layer was larger than that of ceramic layer from cross section microstructure in Figure 9b. The thickness of ceramic layer was approximately 200 μm, and the thickness of pyrolysis layer was approximately 400μm. The formation of multi-layer structure helped to show the material changes better in the ablation process of the composite.

The morphologies of the pyrolysis layer and ceramic layer were shown in Figure 10. Figure 10a shows that a loose structure is formed in the pyrolysis layer. According the analysis of thermal decomposition, lots of gaseous substances were released due to the degradation of silicone rubber to form internal pressure in the material, which produced a large number of pores and cracks. At higher temperature, further reactions among pyrolysis residue were happened to form dense and rigid ceramic for composite with 30 phr ZrSi_2_, as shown in Figure 10b. The surface ceramic structure with a certain mechanical strength can resist the erosion of high-speed airflow, and prevent heat and oxygen from transferring into the composite effectively, which was beneficial to reduce the linear and mass ablation rates of the composite. Figure 10c shows the microstructure of ceramic layer without ZrSi_2_. A large number of holes existed in the ceramic layer. Compared with sample SRZ3, the structure of the ceramic layer was relatively looser. It can be seen from the microstructure with higher magnification that the diameter of the pore was obviously larger. Therefore, the addition of ZrSi_2_ greatly improved the densification of ceramic products during ablation, which formed better protection from airflow erosion and heat diffusion. 

Figure 11 shows the XRD patterns for the ablation layers of the ceramifiable silicone rubber composite. The pyrolysis temperature of silicone rubber was below 800 °C according to the thermal analysis, and the residue was mainly composed of C, SiO_2_ and siliconoxycarbide. Graphite and ZrSi_2_ were relatively stable at below 800 °C, so the main phases of residue in the pyrolysis area were still graphite and ZrSi_2_, as shown in Figure 11a. While the closer the ablation surface was, the higher the temperature was, and the flame temperature of oxyacetylene reached more than 3000 °C. In addition, there were oxygen, water vapor and other oxidative gases in the process of combustion. Therefore, during the oxyacetylene torch ablation, several chemical reactions would happen. The oxidation of graphite can easily take place at high temperature in oxidation environment (Equation (4)). The carbothermal reaction also occurred inside the composite with high temperature due to lack of oxygen, producing SiC (Equation (5)). SiC would also be oxidized into SiO_2_ along with the ablation (Equation (6)). When the ablation temperature was over 1700 °C, a liquid phase was produced by the melting of SiO_2_ (Equation (7)), which prevented the diffusion of oxygen and heat flow to the interior of the composite.
(4)$$\mathrm{Graphite}\;+{\;O}_{2}\;\to {\;\mathrm{CO}}_{2}$$
(5)$$C\;+{\;\mathrm{SiO}}_{2}\;\to \;\mathrm{SiC}\;+\;\mathrm{CO}$$
(6)$$\mathrm{SiC}\;+{\;O}_{2}\;\to {\;\mathrm{SiO}}_{2}\;+{\;\mathrm{CO}}_{2}$$
(7)$${\mathrm{SiO}}_{2}\;\left(s\right)\;\to {\;\mathrm{SiO}}_{2}\;\left(l\right)$$

After cooling, the white residue was formed and adhered to the ablation surface, as shown in Figure 9a. Oxidation reaction of ZrSi_2_ also occurred under the oxyacetylene torch ablation (Equation (8)), and then ZrO_2_ was produced. ZrO_2_ with high melting point has excellent ablation resistance, which has been confirmed in the literature [17]. Furthermore, chemical reaction also took place between ZrO_2_ and SiO_2_ to produce ZrSiO_4_ (Equation (9)). Finally, the C (graphite), SiC and some other products disappeared in the high temperature oxyacetylene flame, while ZrO_2_, SiO_2_ and ZrSiO_4_ were remained to form ceramic structure on the ablation surface. The main phases shown in Figure 11b are ZrO_2_, SiO_2_ and ZrSiO_4_, and the characteristic peaks of the original graphite and ZrSi_2_ disappear.
(8)$${\mathrm{ZrSi}}_{2}\;+{\;O}_{2}\;\to {\;\mathrm{SiO}}_{2}+{\;\mathrm{ZrO}}_{2}$$
(9)$${\mathrm{SiO}}_{2}\;+{\;\mathrm{ZrO}}_{2}\;\to {\;\mathrm{ZrSiO}}_{4}$$

In order to test the mechanical properties of ceramic products, the samples with different content of ZrSi_2_ were heated to 1800 °C in a muffle furnace and held for 15 min. The bending strength of the composite at high temperature was tested. The results are shown in Figure 12. The bending strength of the ceramic products increased with the increasing content of ZrSi_2_. When the content of ZrSi_2_ was 0 phr, the bending strength of the ceramic residue was 5.50 MPa. With the increase of ZrSi_2_ content to 30 phr, the bending strength of the ceramic residue increased to 10.61 MPa. According to the results of phase analysis of ablation residue, ZrO_2_ and SiO_2_ were formed during the ablation process. The SiO_2_ can melt at the ablation temperature and bonded other components together. Moreover, ZrO_2_ would react with SiO_2_ to form new ceramic product. So the bending strength of the residue was increased. The improvement of bending strength indicated that the ceramic products had good mechanical properties to resist erosion of oxyacetylene air flow.

## 4. Conclusions

Based on the analysis of vulcanization, mechanical strength, thermal properties and ablation resistance of the ceramifiable silicone rubber composite, the following conclusions can be drawn about the effects of ZrSi_2_ on the properties of the composite. The introduction of ZrSi_2_ shortened the vulcanization time of the ceramifiable silicone rubber composite. In the FTIR spectrum, the infrared spectrum of samples with different ZrSi_2_ content showed the same characteristic absorption peaks, which showed that ZrSi_2_ did not participate in the reactions of silicone rubber functional groups. The tensile strength decreased with the introduction of ZrSi_2_, but increased first and then decreased with the increased content of ZrSi_2_. The elongation at break decreased, while the permanent deformation increased. When the content of ZrSi_2_ was 30 phr, the tensile strength reached the maximum value of 5.08 MPa, the elongation at break and the permanent deformation were 364.3% and 5.24%, respectively. The thermal conductivity of the composite increased from 0.553 W/(m·K) to 0.694 W/(m·K) as the content of the ZrSi_2_ increased from 0 to 40 phr. In addition, the thermal conductivity of the composite decreased with the increase of temperature. The results of thermal analysis of the composite showed that the addition of ZrSi_2_ increased the initial decomposition temperature of the composite in nitrogen from 473.5 °C to 490.2 °C, while had little effect on the peak temperature of decomposition. However, the thermal decomposition temperature of the composite in air was lower than that in nitrogen. According to the test and calculation of the yield of decomposition residue in nitrogen at 800 °C, the yield of pyrolysis residue was about 50% when the content of ZrSi_2_ was 30 phr. The experimental value was basically consistent with the calculated value. And graphite and ZrSi_2_ had good thermal stability in nitrogen. Nevertheless, in air, when the temperature was over 680 °C, graphite began to oxidize. When the temperature was over 800 °C, ZrSi_2_ began to oxidize and gain weight. The yield of decomposition residue was 45.48% at 1000 °C. The addition of ZrSi_2_ reduced the linear and the mass ablative rate of the composite obviously, improving ablation resistance. With the ZrSi_2_ content of 30 phr, the linear and mass ablation rate were 0.041 mm/s and 0.029 g/s, decreasing by 57.5% and 46.3% compared with the composite without ZrSi_2_, respectively. Consequently, the ceramifiable silicone rubber composite filled with ZrSi_2_ is very promising for TPS.

## Figures and Tables

**Figure 1 polymers-12-00496-f001:**
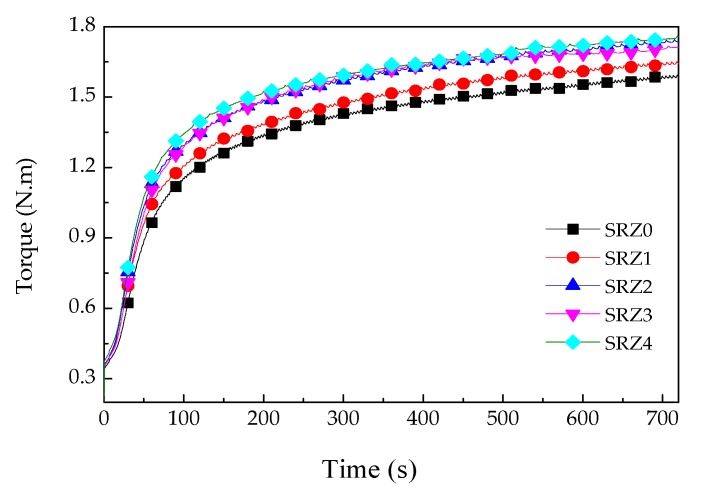
Vulcanization curves of the ceramifiable silicone rubber composites with different content of ZrSi_2_.

**Figure 2 polymers-12-00496-f002:**
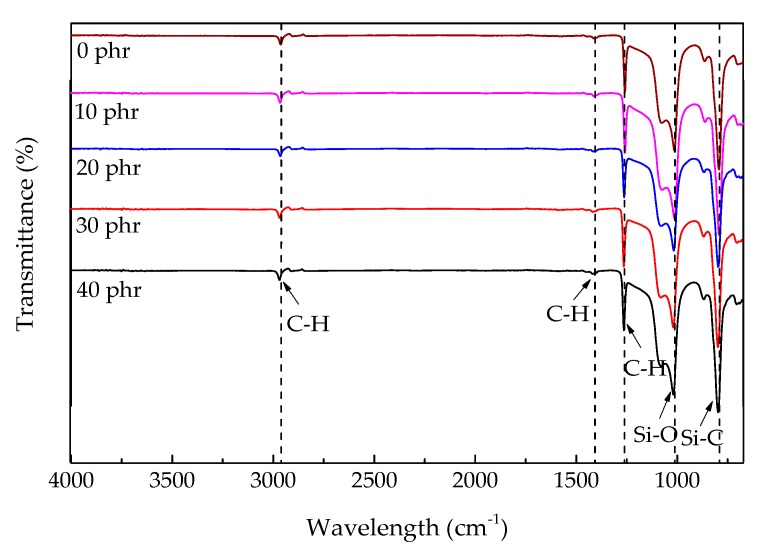
FTIR spectra of the ceramifiable silicone rubber composites.

**Figure 3 polymers-12-00496-f003:**
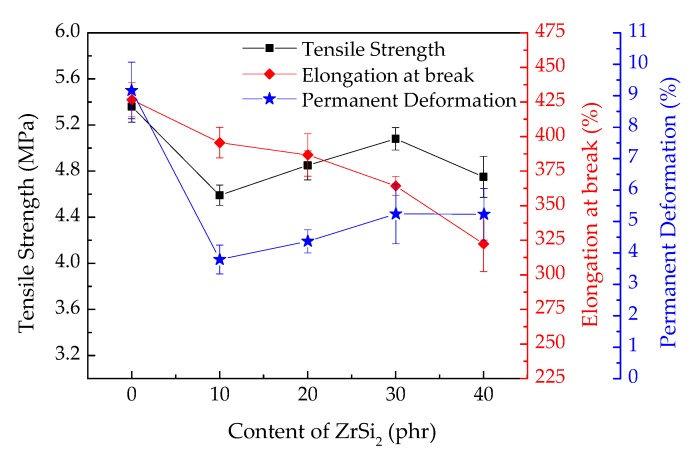
The tensile strength and elongation at break of ceramifiable silicone rubber composites with different content of ZrSi_2_.

**Figure 4 polymers-12-00496-f004:**
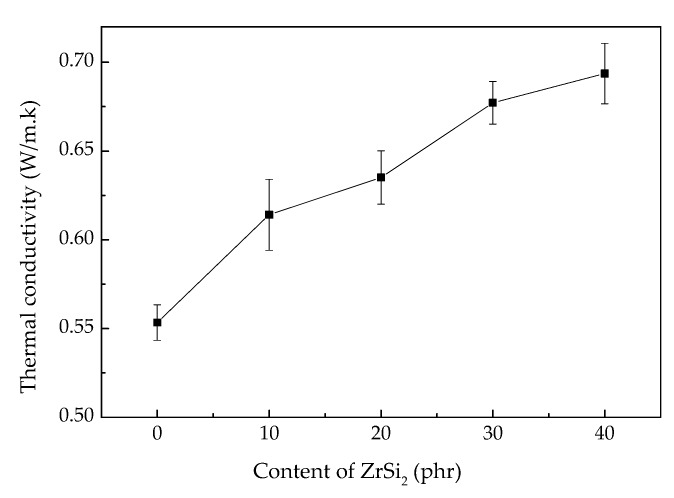
Thermal conductivity of ceramifiable silicone rubber composites with different content of ZrSi_2_.

**Figure 5 polymers-12-00496-f005:**
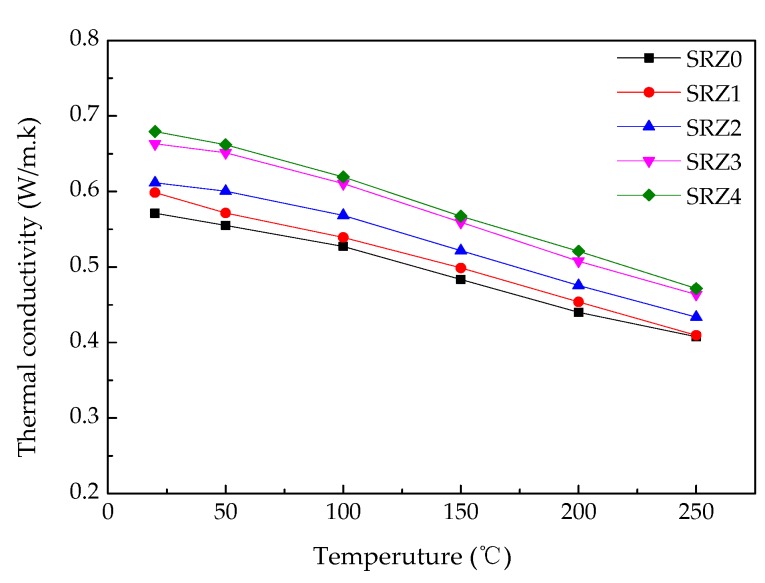
Thermal conductivity of ceramifiable silicone rubber composites with different content of ZrSi_2_ at different temperature.

**Figure 6 polymers-12-00496-f006:**
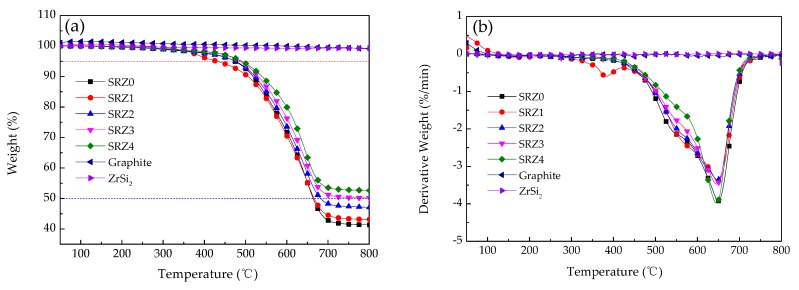
TG (**a**) and DTG (**b**) curves of the ceramifiable silicone rubber composites in N_2_.

**Figure 7 polymers-12-00496-f007:**
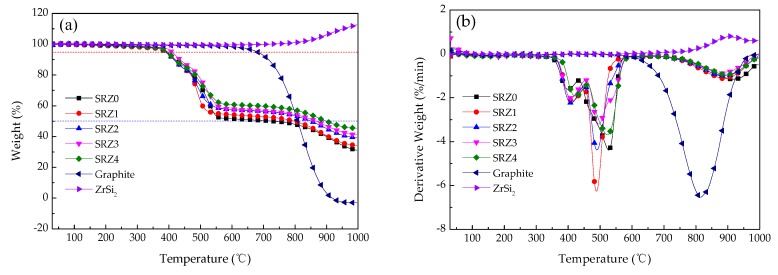
TG (**a**) and DTG (**b**) curves of the ceramifiable silicone rubber composites in air.

**Figure 8 polymers-12-00496-f008:**
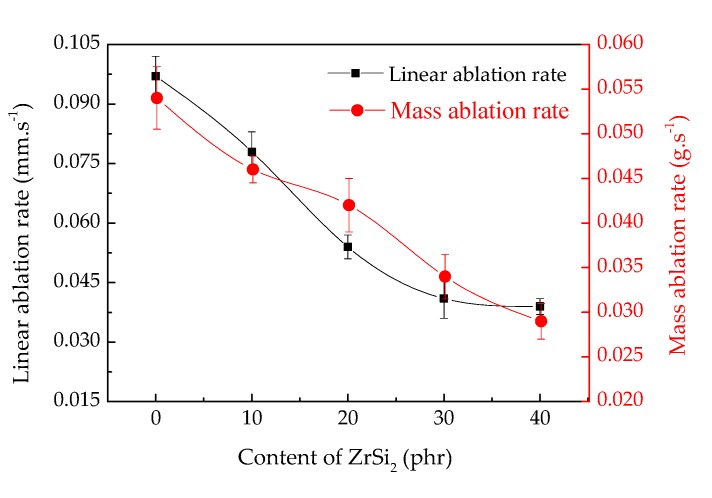
Linear and mass ablation rate of ceramifiable silicone rubber composites with different content of ZrSi_2_.

**Figure 9 polymers-12-00496-f009:**
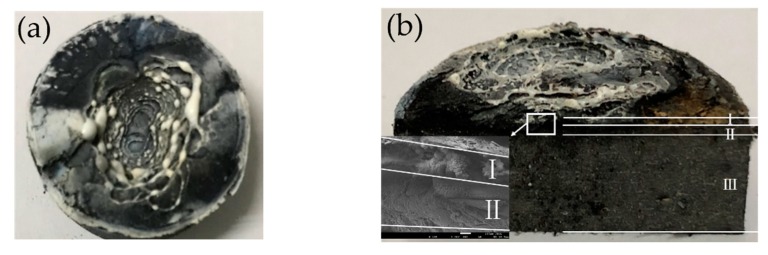
Photographs of ablated composite: (**a**) Surface; (**b**) Cross section.

**Figure 10 polymers-12-00496-f010:**
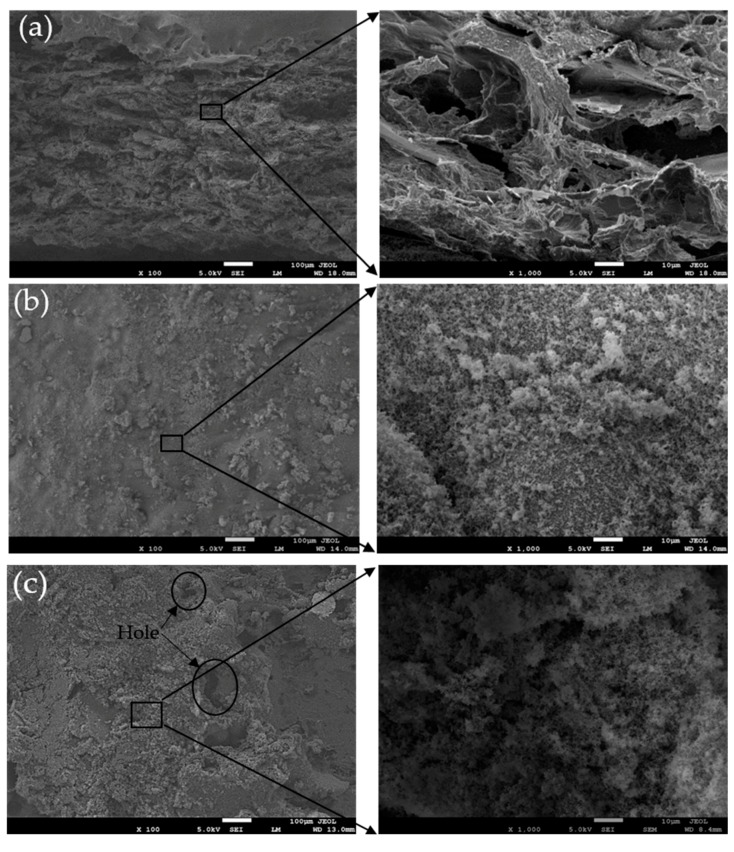
SEM images for the residue of the ceramifiable silicone rubber composite after ablation: (**a**) pyrolysis layer of SRZ3; (**b**) ceramic layer of SRZ3, (**c**) ceramic layer of SRZ0.

**Figure 11 polymers-12-00496-f011:**
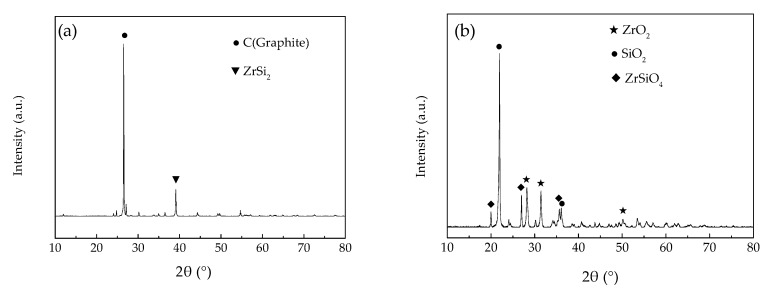
XRD patterns for the ablation area of the ceramifiable silicone rubber composite after ablation (**a**) Pyrolysis layer, (**b**) Ceramic layer.

**Figure 12 polymers-12-00496-f012:**
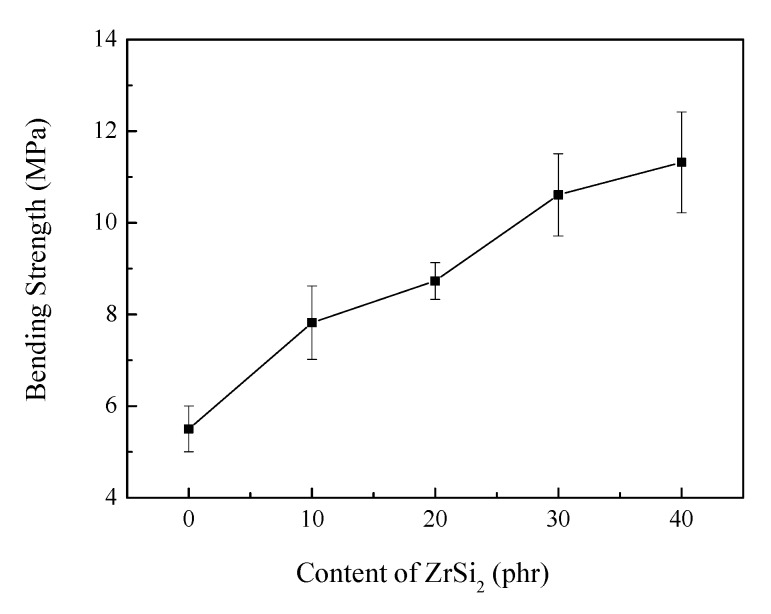
Bending strength of the heat-treatment residue of ceramifiable silicone rubber composites with different content of ZrSi_2_.

**Table 1 polymers-12-00496-t001:** Ingredients (wt. %) of the ceramifiable silicone rubber composites.

Samples	Ingredients in phr ^1^				
	SR	SiO_2_	Graphite	ZrSi_2_	DBPH
SRZ0	100	40	30	0	2
SRZ1	100	40	30	10	2
SRZ2	100	40	30	20	2
SRZ3	100	40	30	30	2
SRZ4	100	40	30	40	2

^1^ Parts per hundreds of silicone rubber.

**Table 2 polymers-12-00496-t002:** Vulcanization characteristic parameters of the composites.

Samples	*T*_10_ (s)	*T*_90_ (s)	Torque (N·m)
SRZ0	16.92	421.80	1.594
SRZ1	16.56	391.68	1.651
SRZ2	16.20	384.48	1.746
SRZ3	16.56	314.76	1.717
SRZ4	16.20	379.80	1.762

**Table 3 polymers-12-00496-t003:** Characteristic decomposition data of the composites in N_2._

Samples	*T*_5_ (°C)	*T_max_* (°C)	Residue (800 °C, %)	Calculated Value (800 °C, %)
SRZ0	473.5	649.0	41.35	41.35
SRZ1	429.9	648.9	43.15	44.54
SRZ2	474.6	647.6	47.08	47.39
SRZ3	486.6	645.1	50.18	49.96
SRZ4	490.2	644.7	52.62	52.29
Graphite	/	/	99.13	/
ZrSi_2_	/	/	99.32	/

**Table 4 polymers-12-00496-t004:** Characteristic decomposition data of the composites in air.

Samples	*T*_5_ (°C)	*T_max_* (°C)	Residue (680 °C, %)	Residue (1000 °C, %)
SRZ0	388.6	527.1	50.64	30.96
SRZ1	392.5	488.0	53.12	34.22
SRZ2	394.6	491.6	54.61	39.32
SRZ3	399.4	499.4	56.83	40.95
SRZ4	399.7	513.7	60.06	45.48
Graphite	680.6	815.1	95.05	0
ZrSi_2_	/	/	99.81	112.92

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
