# Peer review of "Effects of Zirconium Silicide on the Vulcanization, Mechanical and Ablation Resistance Properties of Ceramifiable Silicone Rubber Composites"

_polymers, 2020, doi:10.3390/polym12020496_

Round 1
Reviewer 1 Report
The manuscript (polymers-723084) presented the effect of ZrSi2 on the vulcanization, mechanical and ablation resistance properties of the flexible silicon composite. The work was one of the series report (Materials, 2016, 9, 723). The authors displayed a wide spectrum of data to evidence the positive effect of ZrSi2 in ceramifiable silicon rubber composites. Unluckily, the authors organized these contents in a professional manner. Additionally, the key point to reveal the positive effect ZrSi2 on the vulcanization, mechanical property and ablation resistance were generally attractive to the reviewers as well as the wide readers (Composites Part A: Applied Science and Manufacturing, 2013, 44: 70-77; Plastics, Rubber and Composites, 2016 45: 430-435). The reviewer suggested reconsidering the manuscript after the major revision.
1) After reading through the whole manuscript, the reviewer was impressed by the accurate and abundant characterizations. However, the organization of the contents were unprofessional. The manuscript was written like a technical evaluation report. In the reviewer’s opinion, the authors should focus on the ablation resistance and vulcanization with the other properties the adjuvant parts since either ablation resistance or vulcanization hardly supported one paper. The irrelevant information was moved to supplementary materials. Additionally, The more in-depth data was mandatory to reveal the mechanism. For example, the detailed and comparative SEM observation of the residues was necessary. The in-situ acquisition of the residue for XRD and SEM was at best conducted for clarifying the transformation of different components. The mechanical property of the residue was another crucial parameter for supporting the ablation study.
2) Various fillers have been investigated for ablation reinforcement (Composites Part A: Applied Science and Manufacturing, 2013, 44: 70-77; Plastics, Rubber and Composites, 2016 45: 430-435…….). Unless the use of ZrSi2 gave rise to a remarkable improvement of linear and mass ablation rate compared with the reported data, the manuscript would be interesting.
3) The authors should revise the manuscript as significantly as possible after the coronavirus pneumonia outbreak.
Author Response
Point 1: After reading through the whole manuscript, the reviewer was impressed by the accurate and abundant characterizations. However, the organization of the contents were unprofessional. The manuscript was written like a technical evaluation report. In the reviewer’s opinion, the authors should focus on the ablation resistance and vulcanization with the other properties the adjuvant parts since either ablation resistance or vulcanization hardly supported one paper. The irrelevant information was moved to supplementary materials. Additionally, the more in-depth data was mandatory to reveal the mechanism. For example, the detailed and comparative SEM observation of the residues was necessary. The in-situ acquisition of the residue for XRD and SEM was at best conducted for clarifying the transformation of different components. The mechanical property of the residue was another crucial parameter for supporting the ablation study.

Response 1: Firstly, according to your suggestions, the organization of the contents in this paper was adjusted. The irrelevant experimental results were removed. the micro morphology of the residue of the composite without ZrSi2 after ablation was added in the paper. It was found in the comparison that the number of pores of the ablation residue was reduced after adding ZrSi2, improving the compactness. In addition, in order to test the mechanical properties of ceramifying residue, the samples with different content of ZrSi2 were heated to 1800 °C in muffle furnace and kept for 15 min. The bending strength of the composite at high temperature was tested according to your suggestion. The bending strength of the ceramic residue increased with the increasing content of ZrSi2. However, the in-situ acquisition of the residue for XRD and SEM can not be conducted due to the limitation of experimental conditions. But, we will improve the level of test equipment and conduct research in-depth in the later experiment.
Point 2: Various fillers have been investigated for ablation reinforcement (Composites Part A: Applied Science and Manufacturing, 2013, 44: 70-77; Plastics, Rubber and Composites, 2016 45: 430-435…….). Unless the use of ZrSi2 gave rise to a remarkable improvement of linear and mass ablation rate compared with the reported data, the manuscript would be interesting.
Response 2: Among the inorganic fillers mentioned in the literature (Plastics, Rubber and Composites, 2016 45: 430-435), ablation resistance of the composite is greatly improved by incorporation of ZrB2. The linear ablative rate of the composite filled with ZrB2 is 0.152 mm/s. In the literature (Composites Part A: Applied Science and Manufacturing, 2013, 44: 70-77), The ablation resistance of the composites is greatly improved by incorporation of ZrO2 or ZrC. The composite with 30 phr ZrO2 showed the best ablation properties, whose linear and mass ablation rates are 0.045 mm/s and 0.046 g/s. However, in my study, when the best addition amount of ZrSi2 was 30 phr, the linear and mass ablation rate of the composite were 0.041 mm/s and 0.029 g/s. In addition, short carbon fiber are also introduced into composites to improve ablation resistance in their literature. So the use of ZrSi2 gave rise to a remarkable improvement of linear and mass ablation rate.
Reviewer 2 Report
The undertaken research topic is interesting nevertheless in my opinion presented manuscript (polymers-723084) is not suitable for publication in the Polymers in its present form. The most important issues that rais my reservations are pointed below.
- On page 3 of the manuscript in lines 125-128 the authors write: „Fourier transform infrared spectroscopy (FTIR) was obtained in the range of 400–4000 cm-1 at a 125 resolution of 1 cm-1 on a Nexus FTIR spectrophotometer (Thermo Nicolet, Waltham, MA, USA) using the KBr pellet technique for samples with different content of ZrSi2 to determine whether ZrSi2 participated in the vulcanization of silicone rubber.” The FT-IR spectroscopy could not be obtained. This sentence should be corrected. Moreover, it is not clear for me how the silicone rubber samples were homogenized with KBr. It is rather hard to obtain homogeneous mixtures in this case. In my opinion, it would be better to use another technique like ATR in this case.
- The description of the band at 1412cm-1 (C=H) in Figure 2 is incorrect. It should be C-H.
- I don't understand the sentence on page 5 in line 189: „The addition of ZrSi2 to silicone rubber composite may change the vulcanization network structure of silicone rubber due to the presence of silicon.” What did the authors mean?
- On the basis of results presented in figure 1 (Vulcanization curves) and in Figure 3 (Swelling curves), the authors postulate that the crosslink density of obtained silicone rubbers increases with an increasing amount of ZrSi2. In my opinion, the crosslink density is not affected by the ZrSi2 addition. Observed changes in torque are the result of the physical interaction of filler and polymer matrix. It is also known that swelling decreases with an increasing amount of filler. The swelling ratio of pure rubber should be calculated with a different method (see eg. Fillers and Filled Polymers (Macromolecular Symposia, Book 168), Wiley-VCH, 2001, pp 246-247).
Author Response
Point 1: On page 3 of the manuscript in lines 125-128 the authors write: “Fourier transform infrared spectroscopy (FTIR) was obtained in the range of 400–4000 cm-1 at a 125 resolution of 1 cm-1 on a Nexus FTIR spectrophotometer (Thermo Nicolet, Waltham, MA, USA) using the KBr pellet technique for samples with different content of ZrSi2 to determine whether ZrSi2 participated in the vulcanization of silicone rubber.” The FT-IR spectroscopy could not be obtained. This sentence should be corrected. Moreover, it is not clear for me how the silicone rubber samples were homogenized with KBr. It is rather hard to obtain homogeneous mixtures in this case. In my opinion, it would be better to use another technique like ATR in this case. 

Response 1: I asked the engineer working in the test center about the test method of the thin-film sample, and he told me that the fourier transform infrared spectroscopy was obtained by ATR-FTIR technique at that time. And it has been revised in the paper.
Point 2: 2.The description of the band at 1412cm-1 (C=H) in Figure 2 is incorrect. It should be C-H.
Response 2: It has been revised according to your suggestion in Figure 2.
Point 3: 3.I don't understand the sentence on page 5 in line 189: “The addition of ZrSi2 to silicone rubber composite may change the vulcanization network structure of silicone rubber due to the presence of silicon.” What did the authors mean?
Response 3: Whether ZrSi2 would reinforce silicone rubber like fumed SiO2 is not be conformed. It is also a question for me whether the silicon atoms in the surface of ZrSi2 particles can bond with the molecular chain of silicone rubber. If these reactions can take place, the molecular structure of silicone rubber would be changed. So what I want to express is the effect of ZrSi2 on vulcanization of silicone rubber.
Point 4: On the basis of results presented in figure 1 (Vulcanization curves) and in Figure 3 (Swelling curves), the authors postulate that the crosslink density of obtained silicone rubbers increases with an increasing amount of ZrSi2. In my opinion, the crosslink density is not affected by the ZrSi2 addition. Observed changes in torque are the result of the physical interaction of filler and polymer matrix. It is also known that swelling decreases with an increasing amount of filler. The swelling ratio of pure rubber should be calculated with a different method (see eg. Fillers and Filled Polymers (Macromolecular Symposia, Book 168), Wiley-VCH, 2001, pp 246-247).
Response 4: In this study, it was found that there was physical interaction between ZrSi2 particles and silicone rubber, which did not change the molecular structure of silicone rubber. As you said, the crosslink density is not affected by the ZrSi2 addition. In addition, crosslinking density is also difficult to reflect the vulcanization characteristics of silicone rubber. According to the other reviewer's opinion, the irrelevant information should be moved to supplementary materials. Therefore, this erroneous result was deleted in this paper. But according to your suggestion, I will recalculate the crosslinking density. And the result will be used as an extension of my knowledge.
Round 2
Reviewer 1 Report
The reviewer cautiously checked the revised version. The authors addressed all the comments well. I suggested the acceptance of the manuscript in present form.